# Hierarchically self-assembled hexagonal honeycomb and kagome superlattices of binary 1D colloids

Sung-Hwan Lim[1], Taehoon Lee[1], Younghoon Oh[2], Theyencheri Narayanan [3], Bong June Sung[2] & Sung-Min Choi[1]

Synthesis of binary nanoparticle superlattices has attracted attention for a broad spectrum of potential applications. However, this has remained challenging for one-dimensional nanoparticle systems. In this study, we investigate the packing behavior of one-dimensional nanoparticles of different diameters into a hexagonally packed cylindrical micellar system and demonstrate that binary one-dimensional nanoparticle superlattices of two different symmetries can be obtained by tuning particle diameter and mixing ratios. The hexagonal arrays of one-dimensional nanoparticles are embedded in the honeycomb lattices (for $AB_2$ type) or kagome lattices (for $AB_3$ type) of micellar cylinders. The maximization of free volume entropy is considered as the main driving force for the formation of superlattices, which is well supported by our theoretical free energy calculations. Our approach provides a route for fabricating binary one-dimensional nanoparticle superlattices and may be applicable for inorganic one-dimensional nanoparticle systems.

[1] Department of Nuclear and Quantum Engineering, Korea Advanced Institute of Science and Technology, 291 Daehak-ro, Yuseong-gu, Daejeon 34141, Republic of Korea. [2] Department of Chemistry and Research Institute for Basic Science, Sogang University, 35 Baekbeom-ro, Mapo-gu, Seoul 04107, Republic of Korea. [3] European Synchrotron Radiation Facility, 71 Avenue des Martyrs, F-38043 Grenoble, France. Sung-Hwan Lim, Taehoon Lee and Younghoon Oh contributed equally to this work. Correspondence and requests for materials should be addressed to S.-M.C. (email: sungmin@kaist.ac.kr)

Synthesis of binary or multicomponent nanoparticle super-lattices, which may exhibit new emerging properties through synergetic coupling between different types of nanoparticles, has attracted attention for a broad spectrum of potential applications as well as its own scientific merit[1–10]. The fabrication of binary nanoparticle superlattices with different symmetries has been intensively investigated by using an inter-play of entropy and van der Waals, electrostatic, and other interactions, and the majority of binary nanoparticle superlattices reported so far have been made with spherical nanoparticles with two different sizes[11–15]. For the sterically stabilized colloidal spherical nanoparticle mixtures[16–18], it has been shown that the main driving force for the superlattice formation is the free volume entropy contribution to the free energy. The diameter and number ratios between small and large particles and the volume fraction of particles in solution are the key parameters that determine the symmetry of superlattice. Various binary spherical nanoparticle superlattices with different stoichiometries such as $AB_2$, $AB_5$, $AB_6$, and $AB_{13}$ have been observed experimentally[11]. Very recently, a few different binary superlattices have also been observed in a mixture of spherical nanoparticles and nanorods[19–21] and in a mixture of nanodisks and nanorods[22]. However, systematic experimental studies on the mixtures of two different types of one-dimensional (1D) nanoparticles to form highly ordered binary superlattices have been very rare[23].

Theoretical and experimental studies have shown that the binary colloidal mixtures of 1D particles of the same length but two different diameters[24–32] or of the same diameter but two different lengths[33–36] exhibit a rich phase behavior. However, the reported structures are limited to simple liquid-crystalline structures such as nematic or smectic ordering[37, 38]. The binary mixtures of 1D particles often show an entropically driven demixing transition, separating the two types of 1D nanoparticles into two co-existing subphases, respectively[26, 39, 40]. This inherently prevents the cooperative self-assembling of two types of 1D nanoparticles into highly ordered superlattices. To achieve highly ordered binary superlattices of 1D nanoparticles, therefore, new approaches that can overcome the phase separation of two types of 1D nanoparticles should be designed and explored.

In this study, we investigate how 1D nanoparticles are self-assembled when they are added into a hexagonally packed cylindrical-micellar system, depending on the diameter and number ratios between the 1D nanoparticles and cylindrical micelles. Here, the hexagonally packed cylindrical micellar system can be considered as a pre-formed highly ordered assembly of 1D nanoparticles, which allows us to avoid the phase separation

observed in the simple binary mixture of 1D nanoparticles. The binary mixtures of 1D nano-objects show two types of highly ordered hierarchical superlattices, a hexagonal array of one type of 1D nano-objects embedded in a honeycomb lattice or a kagome lattice of another type of 1D nano-objects, depending on the diameter and number ratios. To understand the thermo-dynamics of the self-assembly, we calculate the free volume and the free energy of the two types of hierarchical superlattices by employing a mean-field theory. The theoretical calculation shows that the formation of hierarchical superlattices can be understood in terms of the free volume entropy-driven particle packing phenomena. This study shows that the binary 1D nanoparticle superlattices with different symmetries can be formed by con-trolling the diameter and number ratios between 1D nano-objects of two different diameters.

## Results

**Synthesis and characterization of 1D nanoparticles ($p$-$C_n$TVB).** Nonionic surfactants penta(ethylene glycol) monododecyl ether ($C_{12}E_5$) in water show an isotropic phase at room temperature and form a hexagonally packed cylindrical-micellar phase at low temperature when the $C_{12}E_5$/water mixing ratio is 45:55 by weight[41], providing an excellent test bed for this study. 1D nanoparticles with different diameters ($p$-$C_n$TVB) were synthe-sized by in-situ polymerization of polymerizable cationic surfac-tants, $n$-alkyltrimethylammonium 4-vinylbenzoate ($C_n$TVB, $n$ is the number of carbon in alkyl chain), which form worm-like micelles in water[42–44]. The diameter of the 1D nanoparticles was controlled by the alkyl chain length ($n$ = 10, 12, 14, 16). The shape and size of $p$-$C_n$TVB were characterized by small-angle neutron scattering (SANS) measurement (Fig. 1a and see the details in Supplementary Note 1). As $n$ is increased from 10 to 16, the diameter of $p$-$C_n$TVB linearly increases from 2.52 to 3.90 nm, making the diameter ratio of $p$-$C_n$TVB to $C_{12}E_5$ cylindrical micelles (which have a diameter of 4.3 nm as explained in the next paragraph) linearly increase from 0.59 to 0.91 (Fig. 1b). This allows us to systematically investigate the effects of diameter ratio of two types of 1D nano-objects on their hierarchical self-assembling structures. The lengths of $p$-$C_n$TVB estimated from SANS data are 39, 23, 29, and 69 nm for $n$ = 10, 12, 14, and 16, respectively.

**SAXS measurements of $p$-$C_n$TVB/$C_{12}E_5$/water.** To understand the self-assembling or packing behavior of 1D nanoparticles with different diameters into the hexagonally packed cylindrical

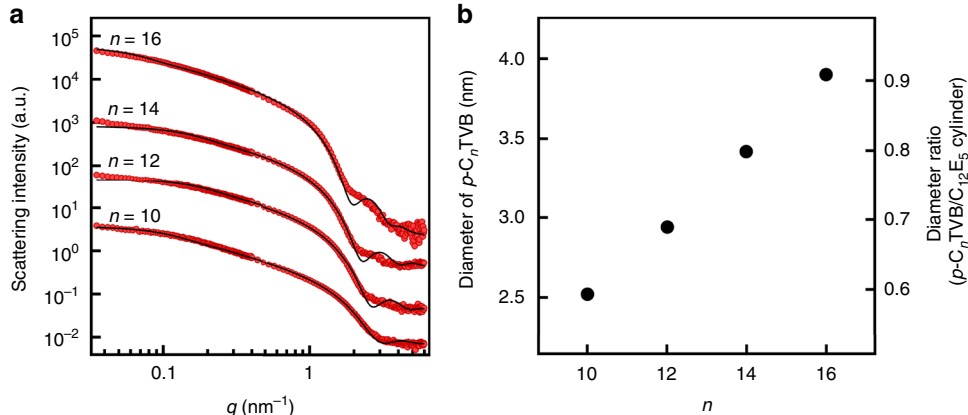

**Fig. 1** SANS form factor analysis of $p$-$C_n$TVB. **a** SANS intensities of 0.1 wt% $p$-$C_n$TVB in $D_2O$ (10 mM NaCl) for $n$ = 10, 12, 14, and 16 (*red*). The *black solid lines* are model fits using a cylindrical particle form factor. $n$ = 10 data are on an absolute scale while other data sets are shifted vertically for visual clarity. **b** The diameters of $p$-$C_n$TVB estimated from the model fittings and the diameter ratios of $p$-$C_n$TVB to $C_{12}E_5$ cylinders

micellar system (Fig. 2), $p$-$C_n$TVB with different $n$ were mixed with $C_{12}E_5$/water (45/55) at the isotropic phase regime, respectively. Here, the weight ratio of $C_{12}E_5$ and water was kept constant at (45/55) and a fixed amount of $p$-$C_n$TVB was added into each sample. The SAXS measurements of $C_{12}E_5$/water (45/55) and $p$-$C_n$TVB/$C_{12}E_5$/water (10/45/55 by weight) for $n = 10, 12, 14,$ and 16 were performed upon cooling from 30 to 4 °C (Fig. 3). Figure 3a shows that $C_{12}E_5$/water (45/55) exists in the isotropic phase at temperature down to 20 °C and, upon further cooling, makes a transition to the highly ordered hexagonal phase as indicated by the sharp scattering peaks with a peak position ratio of $1:\sqrt{3}:2$. The diameter of $C_{12}E_5$ cylinder at the hexagonal phase estimated from the center⁻ to-center distance between the nearest neighboring $C_{12}E_5$ cylinders (5.9 nm) and the volume fraction of $C_{12}E_5$ in the sample (0.48) is 4.3 nm[45], which is larger than all types of $p$-$C_n$TVB used in this study.

The SAXS intensities of $p$-$C_n$TVB/$C_{12}E_5$/water (10/45/55) with different $n$ show two clear features as the mixtures were cooled down from the isotropic to the hexagonal phase (Fig. 3). First, new sharp peaks (as indicated by blue arrows) appear at the low-$q$ region for all the samples, which indicates that a new highly ordered structure with a characteristic length scale larger than

that of $C_{12}E_5$/water (45/55) is formed with the addition of $p$-$C_n$TVB. Second, the phase transition temperature strongly depends on $n$ (i.e. the diameter ratio of $p$-$C_n$TVB and $C_{12}E_5$ cylinder; see Supplementary Note 2).

To understand the structures of $p$-$C_n$TVB/$C_{12}E_5$/water for different $n$ at low temperature, the SAXS intensities of $C_{12}E_5$/water (45/55) and $p$-$C_n$TVB/$C_{12}E_5$/water (10/45/55) at 6 °C are compared (Fig. 4a). The first new peak at $q = 0.75$ nm$^{-1}$, which appears with the addition of $p$-$C_n$TVB, makes a peak position ratio of $1:\sqrt{3}$ with the next peak (which corresponds to the first-order hexagonal peak of $C_{12}E_5$/water (45/55)). Another two new peaks (as indicated by the arrows) also show up making peak position ratios of 2 and $\sqrt{7}$ with respect to the first new peak. This is most clearly visible for $n = 10$ sample. In fact, all the SAXS peaks of each $p$-$C_n$TVB/$C_{12}E_5$/water sample can be indexed by two sets of the same peak position ratio of $1:\sqrt{3}:2:\sqrt{7}:3:\sqrt{12}$ (hexagonal peak ratio), which are directly coupled each other with a scaling of $1:\sqrt{3}$, as indicated by the blue and black numbers. The peaks indexed with black numbers correspond to the scattering peaks of $C_{12}E_5$/water (45/55). These results suggest that at low temperature, all $p$-$C_n$TVB/$C_{12}E_5$/water (10/45/55) samples form a same type of hierarchical hexagonal structure in which a new hexagonal array with a larger lattice parameter is directly coupled with the hexagonal array of $C_{12}E_5$ cylinders. (The peak positions and corresponding lattice parameters of the samples with different $n$ are summarized in Supplementary Table 2.)

To study the effect of mixing ratio of the $p$-$C_n$TVB and $C_{12}E_5$ cylinders, the SAXS intensities of $p$-$C_n$TVB/$C_{12}E_5$/water mixtures at three different concentrations of $p$-$C_n$TVB ($m_p$/45/55, $m_p = 5$, 10, 15) were measured for different $n$. While the SAXS intensities of $n = 12, 14,$ and 16 samples at the different mixing ratios do not show any change in the peak position ratio, the SAXS intensities of $n = 10$ samples show a discrete shift of new peaks to a lower $q$ when $m_p$ is decreased 5 indicating a structural transition (Supplementary Fig. 1). To understand the structural transition in detail, the SAXS intensities of $p$-$C_{10}$TVB/$C_{12}E_5$/water samples with a finer step of $m_p$ variation (5, 6, 7, 8 and 9) were measured

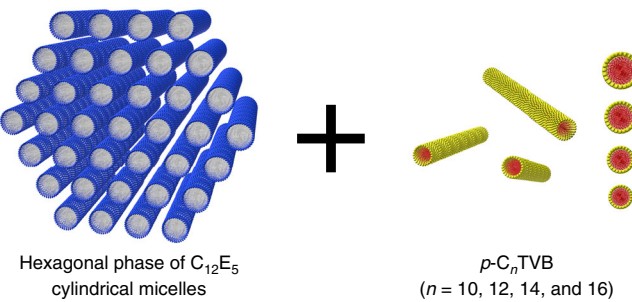

**Fig. 2** Mixing of two types of 1D nano-objects. The cylindrical nanoparticles ($p$-$C_n$TVB, *right*) are added into the hexagonally packed $C_{12}E_5$ cylindrical micellar system (*left*)

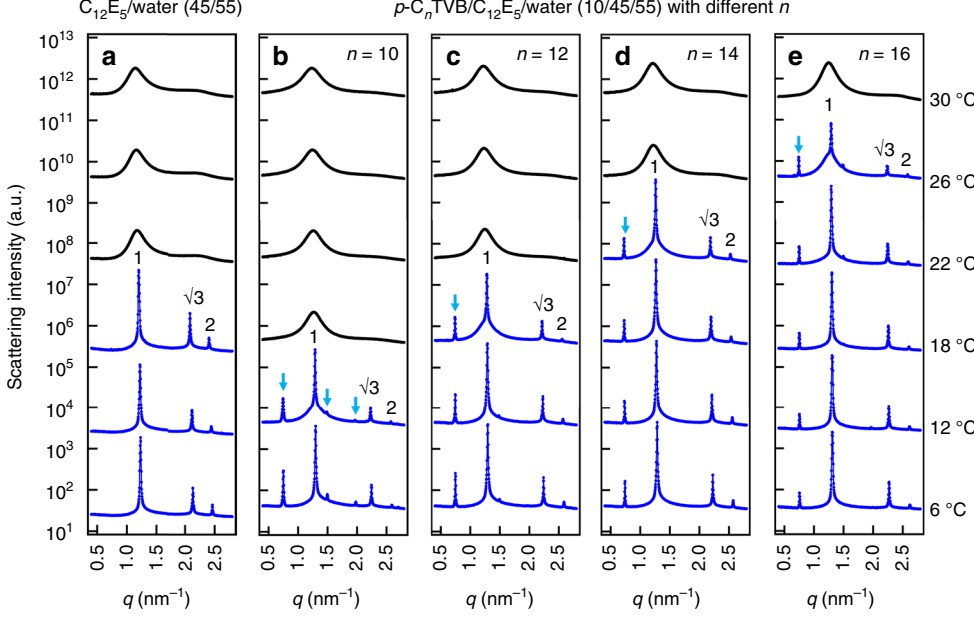

**Fig. 3** Effects of diameter ratio and temperature on the binary mixtures. SAXS intensities of **a** $C_{12}E_5$/water (45/55) and **b**–**e** $p$-$C_n$TVB/$C_{12}E_5$/water (10/45/55) for $n = 10, 12, 14,$ and 16 at different temperatures. All measurements were performed upon cooling down from 30 to 6 °C. Scattering intensities are shifted vertically for visual clarity

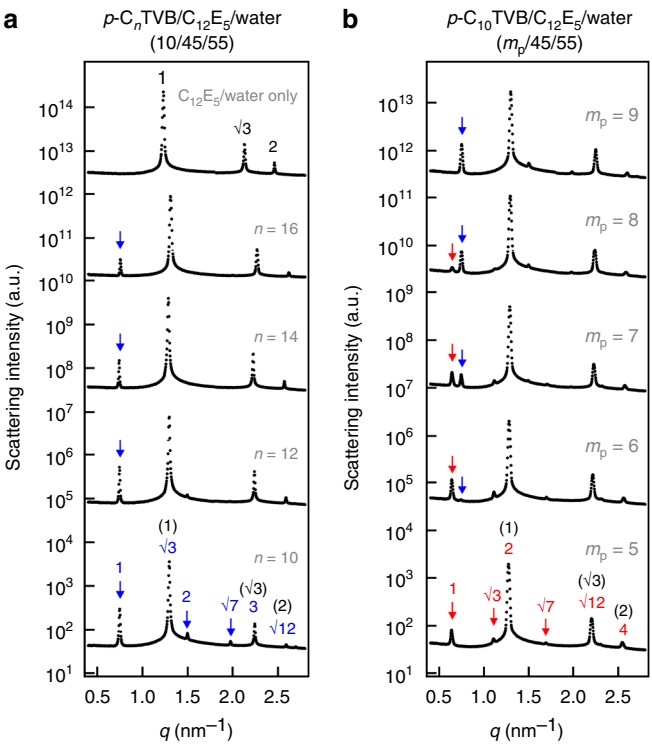

**Fig. 4** Effects of diameter and mixing ratios on the binary mixtures. **a** SAXS intensities of $p$-$C_n$TVB/$C_{12}E_5$/water (10/45/55) for $n = 10, 12, 14$, and 16 in comparison with the SAXS intensity of $C_{12}E_5$/water (45/55). **b** SAXS intensities of $p$-$C_{10}$TVB/$C_{12}E_5$/water ($m_p$/45/55) at different mixing ratios ($m_p = 5, 6, 7, 8$, and 9). All the data were measured at 6°C. Scattering intensities are shifted vertically for visual clarity

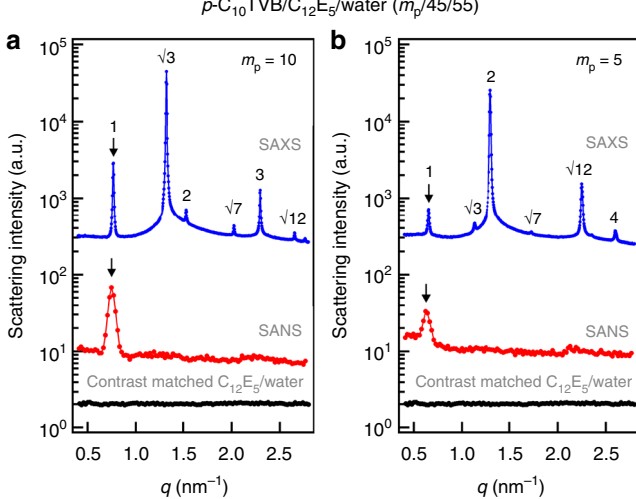

**Fig. 5** Contrast matched SANS measurements. SANS intensities of the partially deuterated $p$-$C_{10}$TVBs mixed with the contrast-matched $C_{12}E_5$/water at mixing ratios of **a** 10/45/55 and **b** 5/45/55. SAXS intensities of each sample are provided for comparison. All the data were measured at 6 °C. Scattering intensities are shifted vertically for visual clarity

(Fig. 4b). As the $m_p$ is decreased from 9 to 5, a new peak shows up at $q = 0.63$ nm$^{-1}$ and grows while the intensity of the peak at $q = 0.75$ nm$^{-1}$ decreases and eventually disappears. Two additional new peaks at $q = 1.10$ and 1.68 nm$^{-1}$ also become clear when $m_p$ is decreased to 5. It should be noted that the new peak at $q = 0.63$ nm$^{-1}$ now makes a peak position of 1:2 with the first-order hexagonal peak of $C_{12}E_5$ cylinders ($q = 1.27$ nm$^{-1}$). (This is in clear contrast with the $p$-$C_{10}$TVB/$C_{12}E_5$/water sample at $m_p = 10$ in which the new peak at $q = 0.75$ nm$^{-1}$ makes a peak position ratio of 1:$\sqrt{3}$ with the first-order hexagonal peak of $C_{12}E_5$ cylinders.) All the SAXS peaks of the $p$-$C_{10}$TVB/$C_{12}E_5$/water at $m_p = 5$ can be indexed by two sets of the same peak position ratio of 1:$\sqrt{3}$:2:$\sqrt{7}$:3:$\sqrt{12}$, which are directly coupled each other with a scaling of 1:2 (indicated by the red and black numbers). This indicates that a discrete structural transition from a hierarchically coupled hexagonal arrays with a scaling of 1:$\sqrt{3}$ to that with a scaling of 1:2 occurs as the concentration of $p$-$C_{10}$TVB is decreased from $m_p = 10$ to $m_p = 5$.

**Contrast matched SANS measurements of $p$-$C_n$TVB/$C_{12}E_5$/water.** Since the new peaks in the hexagonal phase region appear with the addition of $p$-$C_n$TVB while the peaks corresponding to the hexagonally packed $C_{12}E_5$ cylinders are maintained, it is expected that the new peaks arise from the correlations between $p$-$C_n$TVBs. To confirm this, contrast-matched SANS experiments were performed for $p$-$C_n$TVB/$C_{12}E_5$/water mixtures. In this experiment, the neutron scattering length density of $C_{12}E_5$ was matched with that of water by using a mixture of $H_2O$ and $D_2O$ (96:4 by weight) so that $C_{12}E_5$ cylinders in water become invisible by neutrons. To enhance the neutron contrast between $p$-$C_n$TVB and the contrast-matched

$C_{12}E_5$/water, a mixture of deuterated and hydrated $C_n$TVB (1:2 by weight) was used in the synthesis of $p$-$C_n$TVB. This allows us to measure the correlation between $p$-$C_n$TVBs only. The SANS intensity of the contrast-matched $C_{12}E_5$/water (45/55) shows no correlation peak as it should. The SANS intensities of the samples for which partially deuterated $p$-$C_n$TVB are mixed with the contrast-matched $C_{12}E_5$/water (45/55) show a peak that coincides with the first new peak of the SAXS intensities (Fig. 5 for $n = 10$ at (10/45/55) and (5/45/55). Supplementary Fig. 2 for $n = 12, 14, 16$ at (10/45/55)). This confirms that all the new peaks, which appear with the addition of $p$-$C_n$TVBs, originate from the correlation between $p$-$C_n$TVBs.

**Hierarchical binary 1D nanoparticle superlattices.** The SAXS and the contrast-matched SANS results suggest two types of binary superlattices. The $p$-$C_n$TVB/$C_{12}E_5$/water samples ($n = 10, 12, 14$ and 16), of which the SAXS intensity can be indexed with two sets of hexagonal peaks directly coupled each other with a scaling of 1:$\sqrt{3}$, form an AB$_2$-type intercalated hexagonal binary superlattice. Here, a hexagonal lattice of $p$-$C_n$TVBs is embedded in a honeycomb lattice of $C_{12}E_5$ cylinders (Fig. 6a). On the other hand, the $p$-$C_{10}$TVB/$C_{12}E_5$/water sample with a low concentration of $p$-$C_{10}$TVB (5/45/55), of which the SAXS intensity can be indexed with two sets of hexagonal peaks directly coupled each other with $a$ scaling of 1:2, form an AB$_3$ type intercalated hexagonal binary superlattice. Here, a hexagonal lattice of $p$-$C_{10}$TVBs is embedded in a kagome lattice of $C_{12}E_5$ cylinders (Fig. 6b). While the AB$_2$-type intercalated hexagonal structures have been observed in a cationic liposome–DNA complex[46] and a mixture of hydrophilically functionalized single-walled carbon nanotubes and cylindrical surfactant micelles[23], to the best of our knowledge, the AB$_3$-type superstructure observed in $p$-$C_{10}$TVB/$C_{12}E_5$/water (5/45/55) is the first demonstration of the hexagonal lattice embedded in the kagome lattice in binary 1D colloidal systems. The hierarchically coupled binary superstructures of 1D nano-objects are rarely observed in colloidal systems. Especially, the colloidal superstructures which involve the kagome lattice have been reported only for mixture of magnetic and non-magnetic

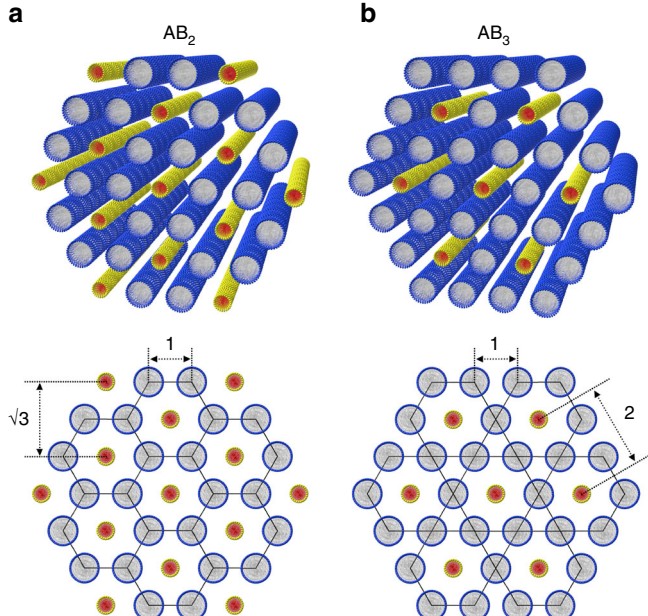

**Fig. 6** Schematics of AB$_2$ and AB$_3$ binary superlattices. Schematics for **a** the AB$_2$ type intercalated hexagonal binary superlattice in which a hexagonal array of $p$-C$_n$TVB is embedded in a honeycomb lattice of C$_{12}$E$_5$ cylinders, and **b** the AB$_3$ type intercalated hexagonal binary superlattice in which a hexagonal lattice of $p$-C$_{10}$TVBs is embedded in a kagome lattice of C$_{12}$E$_5$ cylinders. In the AB$_2$ type structure formed by $p$-C$_{10}$TVB/C$_{12}$E$_5$/water (10/45/55), the distance between the nearest neighboring C$_{12}$E$_5$ cylinders and the distance between the nearest neighboring $p$-C$_{10}$TVBs are 5.60 and 9.71 nm, respectively, making 1:$\sqrt{3}$ ratio. The corresponding distances in the AB$_3$ type structure formed by $p$-C$_{10}$TVB/C$_{12}$E$_5$/water (5/45/55) are 5.72 and 11.44 nm, respectively, making 1:2 ratio

spherical particles under the magnetic field[47], triblock Janus microspheres in water[48], or ditopic molecular bricks on a surface atomic lattice[49].

**SAXS measurements of $p$-C$_n$TVB/C$_{12}$E$_5$/water under shear**. To unambiguously confirm the formation of the intercalated hexagonal structure, additional SAXS measurements for the samples aligned by an oscillatory shear flow[50] were performed. The $p$-C$_{10}$TVB/C$_{12}$E$_5$/water (5/45/55, 10/45/55) and $p$-C$_{14}$TVB/ C$_{12}$E$_5$/water (10/45/55) samples were slowly cooled down from the isotropic phase to the intercalated hexagonal phase under an oscillatory shear with a shear stress of 500 Pa and a frequency of 5 Hz. Once the sample was cooled down into the ordered phase, the shear was stopped and the two-dimensional (2D) SAXS patterns along the radial and tangential directions were measured. In the 2D SAXS patterns measured along the radial direction, all the scattering peaks show up along the direction perpendicular to the flow direction with a narrow azimuthal angular distribution (Supplementary Fig. 3), indicating that the axes of $p$-C$_n$TVB and C$_{12}$E$_5$ cylinders in the samples are unidirectionally aligned parallel to the flow direction. The 2D SAXS patterns measured along the tangential direction show a six-fold symmetry (Fig. 7) with a specific azimuthal angle shift between the neighboring hexagonal patterns as clearly shown in the azimuthally averaged 1D scattering patterns. These results clearly confirm that $p$-C$_n$TVB/C$_{12}$E$_5$/water (10/45/55, $n = 10$, 14) indeed forms the AB$_2$-type intercalated hexagonal structure and $p$-C$_{10}$TVB/C$_{12}$E$_5$/ water (5/45/55) forms the AB$_3$-type intercalated hexagonal structure.

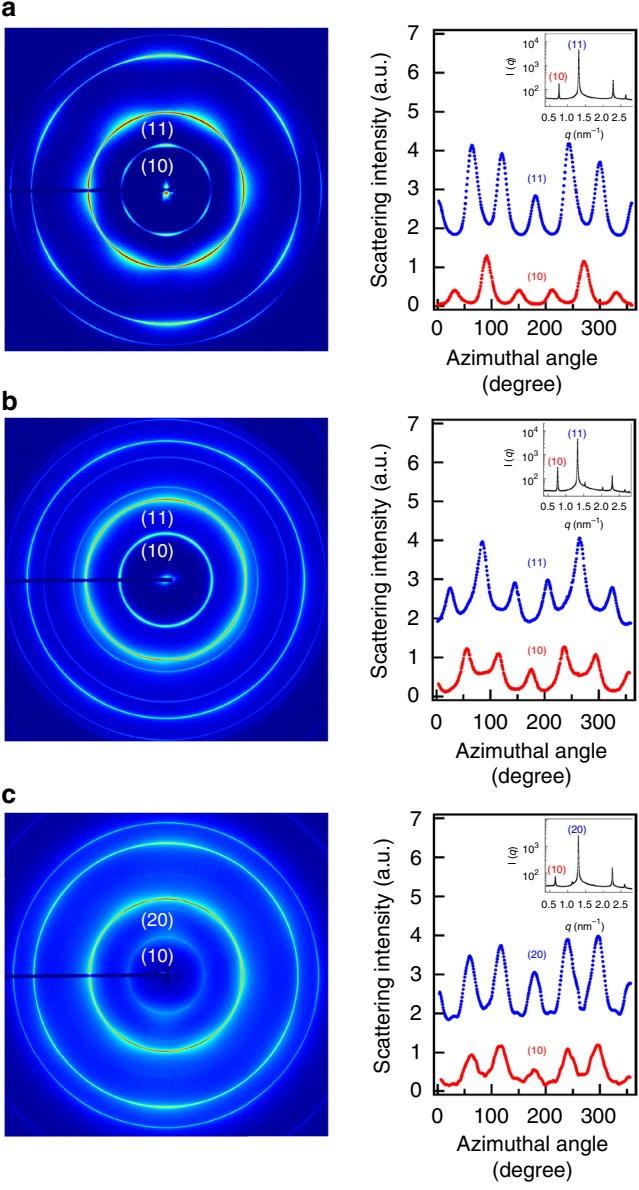

**Fig. 7** Shear-induced alignment of binary superlattices. SAXS measurements along the tangential direction for the samples of **a** $p$-C$_{14}$TVB/C$_{12}$E$_5$/water (10/45/55), **b** $p$-C$_{10}$TVB/C$_{12}$E$_5$/water (10/45/55), and **c** $p$-C$_{10}$TVB/C$_{12}$E$_5$/water (5/45/55). The azimuthally averaged intensities along the (10) and (11) reflections (which make the peak position ratio of 1:$\sqrt{3}$) for AB$_2$ type structure and those along the (10) and (20) reflections (which make the peak position ratio of 1:2) for AB$_3$ type structure are shown on the right together with the 1D SAXS intensities (inset)

**Theoretical calculations for binary 1D nanoparticle superlattice**. Since the interstitial space between C$_{12}$E$_5$ cylinders is about 2.5 nm in diameter, which is smaller than the diameter of $p$-C$_n$TVB for all $n$, $p$-C$_n$TVBs cannot simply go into the interstitial space. Instead, $p$-C$_n$TVBs replace some of the C$_{12}$E$_5$ cylinders in a systematic way, forming hexagonal arrays of $p$-C$_n$TVBs embedded in honeycomb or kagome lattices of C$_{12}$E$_5$ cylinders. It should be noted that the $p$-C$_n$TVBs are charged particles with zeta potentials linearly varying from $-25$ mV (for $n = 10$) to $+13$ mV (for $n = 16$)[43, 44], while C$_{12}$E$_5$ cylinders are not charged. Therefore, the electrostatic

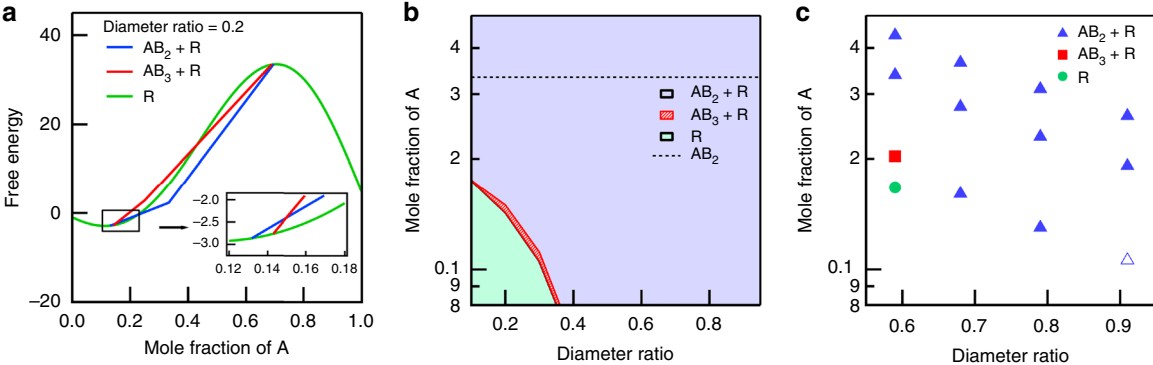

**Fig. 8** Theoretical free energy calculations and phase diagram. **a** Free energies of three different phases ($AB_2+R$ phase (*blue*), $AB_3+R$ phase (*red*), and only R phase (*green*)) for the diameter ratio of 0.2 as a function of the mole fraction of A. **b** Theoretical phase diagram obtained from a mean field theory. Within the $AB_2+R$ phase region, the relative fraction of $AB_2$ and R phases varies with the mole fraction of A and the dotted line indicates the mole fraction of A (0.33) at which the pure $AB_2$ phase is formed without the R phase (Supplementary Fig. 6). **c** Experimental phase diagram determined from the SAXS intensities. At the diameter ratio of 0.91 and the mole fraction of A of 0.11 (an unfilled triangle in **c**), $AB_2+R$ phase is expected from the theory. However, new SAXS peaks corresponding to $AB_2$ superlattice were not observed. This may indicate that the fraction of $AB_2$ superlattce is very low at this point. The variation of integrated intensities of new peaks corresponding to $AB_2$ superlattice with the more fraction of A is given in Supplementary Fig. 7

repulsion between $p$-$C_n$TVBs may play a critical role in the superlattice formation. To identify the effects of electrostatic repulsion, the SAXS measurements with charge screening were performed for $p$-$C_{10}$TVB/$C_{12}E_5$/water, which contains most highly charged particles. The SAXS intensities of $p$-$C_{10}$TVB/$C_{12}E_5$/water (5/45/55 and 10/45/55) in which the electrostatic repulsion was screened by 10 mM NaCl (Supplementary Fig. 4) show the same scattering patterns as those without NaCl (Supplementary Fig. 4). This indicates that the electrostatic repulsion between $p$-$C_n$TVBs is not a main factor in the superlattice formation.

Theoretical and experimental studies have shown that the main driving force for the superlattice formation in mixtures of colloidal spherical particles of two different sizes is free volume entropy, which favors an ordered lattice over a disordered phase to maximize the local free volume available for the spheres, although other energetic interparticle interactions may also be involved[1, 12, 15]. In a binary mixture of hard spheres, the formation of $AB_2$ and $AB_{13}$ superlattices was explained purely by free volume entropy contributions to its free energy[17, 18, 51]. The $AB_{13}$ superlattice was also observed in a binary solution mixture of uncharged block copolymer micelles of two different sizes, which was primarily attributed to free volume entropy contribution as in the hard sphere mixture[52, 53]. The formation of hexagonal arrays of $p$-$C_n$TVBs embedded in honeycomb or kagome lattices of $C_{12}E_5$ cylinders observed in this study can be also understood in terms of free volume entropy contribution.

To estimate the free energy of the mixture of $p$-$C_n$TVBs and $C_{12}E_5$ cylinders, we developed a mean-field theory by considering the free volume of superlattices[51, 54–57]. In our mean-field theory, we model $p$-$C_n$TVBs and $C_{12}E_5$ as two-dimensional hard discs A and B of different effective diameters. The values of effective diameters are tuned in this study such that the interaction between $p$-$C_n$TVBs and $C_{12}E_5$ may be incorporated. Note that mole fractions of A (the number of A discs/total number of discs) are 0.33 and 0.25 in $AB_2$ and $AB_3$ superlattices, respectively. In experiments, however, the mole fraction of A was varied from 0.11 to 0.43 (the mole fraction of $p$-$C_n$TVB is determined by $m_p$, particle diameters, and mass densities, assuming the lengths of $p$-$C_n$TVB and $C_{12}E_5$ cylinders are the same). Therefore, when either $AB_2$ or $AB_3$ superlattice is formed, the rest of A and B particles (that do not participate in the superlattice formation)

are placed on hexagonal lattices in a random fashion, thus forming the random phase. This suggests that $AB_2$ and $AB_3$ superlattices should coexist with a random phase. Therefore, we calculate the partition functions (the free volumes), chemical potentials and free energies per site of superlattices and the random phase, and construct a phase diagram (Fig. 8). More details of our mean-field theory are described in Supplementary Note 3.

Figure 8a depicts the theoretical results for the free energy for all three phases: $AB_2 + R$ ($AB_2$ superlattice + random coexisting phase), $AB_3 + R$ ($AB_3$ superlattice + random coexisting phase), and R (random phase) for the diameter ratio of 0.2 (diameter of disc A/diameter of disc B). For small (or large) mole fraction of A, only R phase may appear because the fraction of A (or B) discs is not high enough to form superlattices. For intermediate values of mole fraction of A between 0.13 and 0.69, R phase coexists with superlattices. For most cases, the $AB_2 + R$ phase has the lowest free energy such that the $AB_2 + R$ phase is considered as a thermodynamically stable phase. Only when the mole fraction of A is close to 0.145, the free energy of $AB_3 + R$ phase is lower than that of $AB_2 + R$, which suggests that the $AB_3 + R$ phase should be more stable (the inset of Fig. 8a). The free energy of R phase is, however, still the lowest around the mole fraction of A of 0.145. Therefore, if $AB_3 + R$ phase were to be observed in experiments, our mean-field theory suggests that the $AB_3 + R$ phase should be a metastable phase. The metastable $AB_3 + R$ phase is predicted by our theory only for small values of diameter ratio. Such a discrepancy between experiments and a theory was also reported for the superlattice formation of a binary hard-sphere mixture[17]. While the theory predicted a solid A + fluid F phase for the binary hard-sphere mixture, the solid $AB_2$ + fluid F phase was observed in experiments. It was also attributed to the fact that solid $AB_2$ + fluid F might be a metastable state.

Figure 8b depicts the phase diagram as a function of the diameter ratio of A to B discs and the mole fraction of A. When both the diameter ratio and the mole fraction of A is small, the R phase may appear. Only around the phase boundary between the R phase and $AB_2 + R$ phase, metastable $AB_3 + R$ phases may exist. The theoretical phase diagram is qualitatively consistent with our experiments (Fig. 8c). In experiments, for most cases, the $AB_2 + R$ phase is observed, which is also expected from our theoretical phase diagram. More interesting is that for small values of the diameter ratio and the mole fraction of A, only R phase is

observed in experiments, which is also consistent with the theory. $AB_3 + R$ is observed in experiments only when both the diameter ratio and the mole fraction of A are relatively small. In the theoretical phase diagram, we can find a metastable $AB_3 + R$ when both the diameter ratio and the mole fraction of A are relatively small. Even though our theory reproduces experiments qualitatively well, it is still based on the mean-field approximation, which ignores the spatial correlation between A and B discs on the hexagonal lattice. It would be, therefore, interesting to incorporate the correlation into the calculation of the free energy. However, it should require accurate information on the interparticle interaction between A and B discs, which is not usually known a priori. Such information might be provided by extensive atomistic molecular dynamics simulations, which would be performed in our future study.

Likos and Henley[58] performed theoretical investigation for a binary hard disc mixture of different sizes at a zero temperature limit, thus ignoring any entropic contribution to the free energy. Setting up a Hamiltonian as the product of the system volume and pressure, they considered various structures of the binary hard disc mixtures and constructed a phase diagram as a function of the disc size ratio and the number fraction. Their theory is different from ours in the sense that we incorporate the free volume entropy into the free energy. However, both their theory and ours suggest that the binary hard disc mixtures form pure phases at specific mixing ratios but mostly form coexisting phases in the similar size ratio and number fraction investigated in our study, although the detailed nature of coexistence is different from each other due to temperature or entropic effects.

In summary, we investigated the packing behavior of 1D nanoparticles ($p$-$C_n$TVBs) of different diameters into a hexagonally packed cylindrical micellar system ($C_{12}E_5$/water). Two different types of binary superlattices, the hexagonal arrays of $p$-$C_n$TVBs embedded in honeycomb lattices of $C_{12}E_5$ cylinders ($AB_2$ type) and the hexagonal arrays of $p$-$C_n$TVBs embedded in kagome lattices of $C_{12}E_5$ cylinders ($AB_3$ type), were observed depending on the particle diameter and mixing ratios. This study shows that the binary supperlattices of 1D nano-objects of different symmetries can be obtained by tuning the particle diameter and mixing ratios. It was understood that the formation of $AB_2$ and $AB_3$ superlattices are driven by free volume entropy maximization. Our theoretical free energy calculations of binary disc superlattices based on a mean field theory which considers the free volume of the superlattices are qualitatively consistent with our experimental results. The understanding obtained in this study may provide new insights for designing synthesis methods for highly ordered metallic, semiconducting or magnetic 1D nanoparticle binary superlattices of different symmetries and functionalities, which are of great current interests for possible applications in photovoltaics, catalysts, and molecular sensing[7, 59, 60].

## Methods

**Materials**. Decyltrimethylammonium bromide ($C_{10}$TAB), dodecyltrimethylammonium bromide ($C_{12}$TAB), myristyltrimethylammonium bromide ($C_{14}$TAB), and hexadecyltrimethylammonium bromide ($C_{16}$TAB) were purchased from Sigma-Aldrich. Deuterated $C_n$TABs ($n = 10, 12, 14,$ and 16), of which hydrogens in alkyl chain are substituted by deuterium, were purchased from C/D/N Isotopes Inc. Dowex Monosphere 550A (OH) anion exchange resin and 4-vinylbenzoic acid (VBA) were purchased from Sigma-Aldrich. The water soluble free-radical initiator, 2,2′-azobis(2-(2-imidazolin-2-yl)propane) dihydrochloride (VA-044), was purchased from Wako Chemicals. Pentaethylene glycol monododecyl ether ($C_{12}E_5$) was purchased from Tokyo Chemical Industry. $D_2O$ (99.9 mol% deuterium enriched) was purchased from Cambridge Isotope Laboratory. All materials were used as received. $H_2O$ was purified using a Millipore Direct Q system (electrical resistivity 18.2 MΩ) prior to use.

**Synthesis of $p$-$C_n$TVB**. $p$-$C_n$TVBs were prepared as described elsewhere[42–44]. Briefly, $n$-alkyltrimethylammonium hydroxide ($C_n$TAOH, $n = 10, 12, 14,$ and 16) was synthesized by replacing $Br^-$ in $C_n$TAB with $OH^-$ using anion exchange resin. $n$-Alkyltrimethylammonium 4-vinylbenzoate ($C_n$TVB) was synthesized by a neutralization of $C_n$TAOH with same stoichiometric amount of VBA followed by repeated crystallizations. Here, hydrated and deuterated $C_n$TVB, which were used to control the neutron scattering contrast, were synthesized from hydrated and deuterated $C_n$TAB, respectively. $p$-$C_n$TVBs were synthesized through in situ free radical polymerization of the counterions ($VB^-$) of $C_n$TVBs which form cylindrical micelles in water. For the polymerization, the concentrations of $C_n$TVB in water were chosen as 1 wt% for $n = 12, 14,$ and 16 and 3 wt% for $n = 10$, considering the critical micellar concentration of each surfactant.

**Preparation of $p$-$C_n$TVB/$C_{12}E_5$/water mixtures**. Binary mixtures of $p$-$C_n$TVB/$C_{12}E_5$ cylinders with different mixing ratios were prepared by adding different amounts of $p$-$C_n$TVB into $C_{12}E_5$/water (45/55 by weight) solution. For homogeneous mixing, the samples were vortex-mixed, and then centrifuged more than 100 times in alternating directions at the isotropic phase. All the samples were stabilized for a day at room temperature before all scattering measurements.

**Small-angle X-ray scattering measurements**. SAXS measurements were performed at the beamline 4C at the Pohang Accelerator Laboratory (PAL), Republic of Korea. X-rays of a wavelength ($\lambda$) of 0.12 nm and a wavelength spread ($\Delta\lambda/\lambda$) of $2 \times 10^{-4}$ delivered by an Si(111) double crystal monochromator were used. A 2D CCD camera (SX165; Mar USA, Inc. CCD 165) was used to collect scattered X-rays. The sample-to-detector distance of 1 m was used to cover the $q$ range of 0.4 $nm^{-1} < q < 3.7$ $nm^{-1}$, where $q = (4\pi/\lambda)\sin(\theta/2)$ is the magnitude of the scattering vector and $\theta$ is the scattering angle. The $q$-values were calibrated using silver behenate ($AgO_2C(CH_2)_{20}CH_3$). Sample cells with a path length of 0.8 mm sealed by two thin Kapton films were used. Temperature was controlled using a water circulation bath (Lauda, Germany). All the samples were stabilized for 20 min at each temperature.

**SANS measurements**. SANS measurements were performed using the 40 m SANS instrument at HANARO, the Korea Atomic Energy Research Institute (KAERI) in Daejeon, Republic of Korea. Neutrons with $\lambda$ of 0.6 nm and $\Delta\lambda/\lambda$ of 0.12 were used. Two sample-to-detector distances of 2.5 and 15.89 m were used to cover the $q$ range of 0.04 $nm^{-1} < q < 4$ $nm^{-1}$. Measured scattering intensities were corrected with background scattering, empty cell scattering, and the sensitivity of individual detector pixels. The corrected data were placed on an absolute scale through the direct beam flux method. The quartz sample cells of 1 and 4 mm path lengths were used for contrast-matched SANS measurements and particle form factor measurements, respectively. Temperature was controlled by using a circulation bath (Neslab RTE 740, USA).

**SAXS measurements under an oscillatory shear flow**. The SAXS measurements under an oscillatory shear flow were performed at the beamline ID02 at the European Synchrotron Radiation Facility (ESRF) in Grenoble, France. X-rays with $\lambda$ of 0.1 nm and $\Delta\lambda/\lambda$ of $2 \times 10^{-4}$ were used. A high-resolution CCD camera (Rayonix MX170 CCD) was used to collect scattered X-rays. The sample-to-detector distance of 1.2 m was used to cover the $q$ range of 0.1 $nm^{-1} < q < 2.9$ $nm^{-1}$. For the oscillatory shear, a Haake RS6000 stress-controlled rheometer with a Couette shear cell with a gap of 1 mm between the inner rotor (20 mm in diameter and 40 mm in height) and the outer stator (22 mm in diameter) was used[50]. The wall thickness of Couette shear cell around the beam position is 100 μm. To see the intensity oscillation in the azimuthally averaged data more clearly, the scattering intensities along the $q_y$ direction (azimuthal angles of 90° and 270°), which appear more strongly in tangential measurements for highly oriented 1D objects, were corrected (for data in Fig. 7c only).

**Zeta potential measurements**. Zeta potential measurements were carried out using a ZetaPlus zeta potential analyzer (Brookhaven Instruments Corporation). The zeta potentials of $p$-$C_n$TVBs were calculated from the measured electrophoretic mobilities of $p$-$C_n$TVBs using Smoluchowski's equation. The zeta potentials of $p$-$C_n$TVBs are −25, −9, 6, and 13 mV for $n = 10, 12, 14$ and 16, respectively. The difference in zeta potentials can be attributed to the difference in the relative number of counterions adsorbed (and polymerized) on the surface of cylindrical micelles, which depends on the critical micellar concentration of each $C_n$TVB[43, 44].

**Data availability**. The authors declare that all data supporting the findings of this study are available within the manuscript and the Supplementary Information or from the corresponding author upon request.

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

## Acknowledgements

This work was supported by the NRF grants funded by the MEST of the Korean government (No. 2017R1A2A1A05001425 and 2011-0031931) and the KAERI grant. We acknowledge the HANARO Neutron Research Center (40m SANS), the PAL (beamline 4C), and the ESRF (beamline ID02) for providing access to the beamlines used in this study.

## Author contributions

S.-H.L., T.L. and S.-M.C. conceived the study. S.-H.L. and T.L. prepared the samples and performed SAXS and SANS experiments and data analysis with guidance of S.-M.C. B.J.S. and S.M.C. conceived the theoretical calculation, which Y.O. and S.-H.L. performed. T.N. co-designed and co-performed SAXS experiment under the shear flow. S.-H.L., T.L., Y.O., and S.-M.C. wrote the manuscript with comments and edits contributed by all the authors.

## Additional information

**Competing interests:** The authors declare no competing financial interests.

