## [Peer Review File · Nature Communications]

Reviewer #1 (Remarks to the Author):

Reviewer's comments

In this paper, the authors experimentally demonstrate for the first time binary 1D nanoparticle superlattices of two different symmetries (AB₂ and AB₃ types) by tuning particle diameter and mixing ratios. They have used scattering measurements (SANS and SAXS) and theoretical calculations to explore the binary 1D nanoparticle superlattices. It also explores the phase diagram of binary mixtures of 1D nanoparticles. The experiments are carefully done, and the message is conveyed in a clear manner. The current study produces a new aspect to the scientific literature and also appeals to the wider audience. However many issues need to be addressed in order to get published in Nature Communications.

Detailed comments

- The authors should refer to "Complex alloy phases for binary hard-disk mixtures", by Christos N. Likos and C. L. Henley, *Philos. Mag. B.* 68, 85-113 (1993) where they investigated the two dimensional binary mixture of hard discs. Author should compare their results with this work. Should cite the work of Klapp et al. (*J. Phys.: Condens. Matter* 28 (2016) 244022) for the rheological measurements.
- Authors have not mentioned in the paper at what pH they performed the measurements.
- Fig2. The curve for the data n=16 doesn't fit to the second part of the experimental data but there is no explanation for this behavior.
- To me it seems most of the SAXS profiles are shifted vertically for clarity, since the talk about the random coexistence phase along with the binary superlattices, I suggest authors to plot the intensity or the area under the peaks as a function of temperature of mixing ratio in order to get a clear picture of the coexistence and it may also provide a qualitative picture.
- It may be speculative to understand the phase behavior only from scattering measurements; It would be useful to perform another independent experiment with Cryo-TEM but not mandatory.
- Line 275, authors claim that the electrostatic repulsion between p-CnTVBs is not the main factor in the superlattice formation. In fig.4 they observed all the possible peaks for n=10 in the mixture and for n=10, zeta potential is -25 mV. Did the authors check their claim that there are no charging effects on C12E5 cylinders by screening the interactions with salt for all the mixing ratios and n? What is the effect of pH?

Reviewer #2 (Remarks to the Author):

The manuscript of Liam et. al. describes periodic assemblies from binary 1D micelle-like structures. In my opinion, the authors demonstrate the interesting cases of assembly that are to the best of my knowledge were not discussed before, that is important from fundamental point of view. The experimental work is detailed and analysis seems to be thorough. I can see this manuscript being published in Nature Communication. The only issue that I want to bring up is that the text of the manuscript is overwhelmed by technical details - it is well written text, but since it has some technical details it is hard to follow the idea of the experiment and the conclusions. Also it will be great if authors will add the comments about the practical value of this study.

Reviewers' comments:

Reviewer #1 (Remarks to the Author):

In this paper, the authors experimentally demonstrate for the first time binary 1D nanoparticle superlattices of two different symmetries (AB₂ and AB₃ types) by tuning particle diameter and mixing ratios. They have used scattering measurements (SANS and SAXS) and theoretical calculations to explore the binary 1D nanoparticle superlattices. It also explores the phase diagram of binary mixtures of 1D nanoparticles. The experiments are carefully done, and the message is conveyed in a clear manner. The current study produces a new aspect to the scientific literature and also appeals to the wider audience. However many issues need to be addressed in order to get published in Nature Communications.

Detailed comments

The authors should refer to "Complex alloy phases for binary hard-disk mixtures", by Christos N. Likos and C. L. Henley, *Philos. Mag. B.* 68, 85-113 (1993) where they investigated the two dimensional binary mixture of hard discs. Author should compare their results with this work. Should cite the work of Klapp et al. (*J. Phys.: Condens. Matter* 28 (2016) 244022) for the rheological measurements.

[Reply]

a) First of all, we would like to thank Reviewer 1 for informing us important references. As suggested by the reviewer, we cited the work by Likos and Henley [reference 58, *Philos. Mag. B.* 68, 86 (1993)] and compared with our results in the revised manuscript. Likos and Henley conducted a systematic investigation on the phase diagram of 2D binary mixtures of hard discs of different sizes. They considered a system in contact with a large pressure bath at a zero temperature limit and set up a Hamiltonian (H) of the system as $H = PV$, where P and V denote the pressure and volume of the system, respectively. Since they were interested in a zero temperature limit, they ignored the entropic contributions to the free energy as they mentioned. They carried out theoretical investigation on many different structures of 2D binary mixture and constructed a phase diagram as a function of the size ratio and the number fraction. Their approach differs from ours in the sense that we considered entropic contribution to the free energy: in our approach the free area for each hard disc is calculated to obtain the free energy. However, both their theory and ours suggest that the binary hard disc mixtures mostly form coexisting phases in the similar size ratio and number fraction investigated in our study, although the detailed nature of coexistence are different each other due to temperature or entropic effects (considered in our theory). While their theory suggests coexistence of two distinct phases (for example, AB₂+B), our theory suggests that either AB₂ or AB₃ hexagonal phase should coexist with a random phase. We add the above discussion to the new version of the manuscript in order to comply with Reviewer 1. (page 19, line 349)

b) Klapp et al (*J. Phys.: Condens. Matter* 28 (2016) 244022) performed theoretical investigations on the orientational dynamics of binary mixtures of rod-like particles of the same diameter but different length under Couette shear flow, and found a variety of

dynamical states. As suggested by the reviewer, we cited the paper in the revised manuscript as reference [36] (page 2, line 52)

Authors have not mentioned in the paper at what pH they performed the measurements.

[Reply]

As it is stated in Materials section, water purified using a Millipore Direct Q system (electrical resistivity 18.2 M Ω) is used in all our experiment (neutral pH, without any further control of pH).

Fig2. The curve for the data $n=16$ doesn't fit to the second part of the experimental data but there is no explanation for this behavior.

[Reply]

We thank the reviewer for pointing out this. We realized that the polydispersity of particle diameter was not considered in the SANS form factor model fitting of $n = 16$ data, while it was considered for all other data. This made the difference for $n = 16$ data. We performed the SANS model fitting again for $n = 16$ data considering the polydispersity of particle diameter. Fig. 2 and Supplementary Table 1 (which summarizes the fitting parameters) have been revised accordingly. (page 5, line 103)

To me it seems most of the SAXS profiles are shifted vertically for clarity, since they talk about the random coexistence phase along with the binary superlattices, I suggest authors to plot the intensity or the area under the peaks as a function of temperature or mixing ratio in order to get a clear picture of the coexistence and it may also provide a qualitative picture.

[Reply] We thank the reviewer for insightful comments. When we tried to answer to this comment, we realized that the relative fractions of the binary superlattice and random phases (which are calculated from our theory, as shown below) were not included in the manuscript and supplementary information. These are added in the revised ones. We also revised Fig. 8b to include the pure AB₂ phase line (the mole fraction of A = 0.33) which was missing in our previous manuscript. (page 17, line 304)

As suggested by the reviewer, we made plots as following,

Plot A: Integrated intensities of the 1st new peak ($q=0.75\text{nm}^{-1}$) vs. mixing ratio
(at a fixed temperature and for all different n)

Plot B: Normalized integrated intensities of the 1st new peak ($q=0.75\text{nm}^{-1}$) vs. temperature
(at each mixing ratio and n)

In the scattering experiments, the fraction of binary superlattice phase relative to the random phase would be reflected in the integrated intensity of the scattering peaks which originate purely from the binary superlattices. For the AB₂ superlattice, the new peak at $q=0.75\text{nm}^{-1}$

comes purely from the binary superlattices and used for the plot a). Since AB_3 superlattice is formed only in $n=10$ at a very narrow mixing ratio range, the scattering intensities corresponding to AB_3 phase is not included in the plot.

Since there is no peaks which originates purely from the random phase (all the peaks other than new peaks are contributed from both the binary superlattice and random phases), it is not feasible to directly identifying the variation of random phase fraction with mixing ratio or temperature.

Plot A shows that the integrated intensities of the 1st new peak monotonically increase with the mole fraction of A until the mole fraction of A becomes 0.33, indicating that the fraction of AB_2 superlattice relative to the random phase increases. This is consistent with the theoretically calculated fraction of the AB_2 phase as shown in Supplementary Fig. 6. Above 0.33, the increase of intensity is significantly suppressed or saturated. This change in intensity variation across 0.33 is consistent with the theoretical calculation although the measured intensity does not show the decrease in intensity as expected from the theoretical calculation. These are included in the manuscript and supplementary information.

Plot B

Plot B shows the normalized integrated intensities of the 1st new peak ($q=0.75\text{nm}^{-1}$, for AB_2 superlattice) vs. temperature (at each mixing ratio and n). Here, the normalization was made with the intensity at the lowest temperature (6 °C). The integrated intensities remain more or less constant with temperature (except near the transition temperature to the isotropic phase). This may indicate that the relative fraction of AB_2 phase remain rather constant within the temperature range (which is not so wide) measured in this study.

However, the data for (5/45/55) with $n=14$ and (10/45/55) with $n=10$ and 16 show rather inconsistent behavior (either decrease or increase). This may be attributed to the possible fluctuation of scattering intensities due to the change of crystallite orientation during measurements. As shown below, the 2D SAXS pattern of static sample shows rings which consist of strong spots coming from crystallites which satisfy Bragg condition at different orientations. Therefore, the integrated peak intensity would depend on the number of crystallites that satisfy the Bragg condition or the degree of satisfaction, which may change during measurements.

Considering this, we think that the reliable use of the integrated peak intensity data for detailed discussion is rather limited, especially for temperature variation experiment (for which rather smaller variation of intensity is expected compared with the mixing ratio variation experiment), although the data still show the overall trend. Therefore, we prefer not to include the peak intensity variation with temperature in the manuscript.

It may be speculative to understand the phase behavior only from scattering measurements; It would be useful to perform another independent experiment with Cryo-TEM but not mandatory.

[Reply]

We agree with the reviewer that real space images such as high resolution cryo-TEM could have been beneficial for our paper. In fact, we tried to measure cryo-TEM images of our samples. However, we were not able to identify the nanoscale structures in our samples. We were told that within the capacity of technical expertise of cryo-TEM personnels at KAIST and KBSI (Korea Basic Science Institute) we worked with, it is not quite feasible to identify the structures in our samples. Especially, we were told that the high viscosity of our sample makes it difficult to prepare thin reliable TEM samples. While real space images could have further confirmed our conclusions, we think that our SAXS (static and shear aligned) and contrast matched SANS analysis strongly support our conclusions on the structure of binary superlattice. If real space images are absolutely required for our manuscript to be accepted, we can try to measure again. However, we hope that the reviewer understand the technical difficulties we are facing.

Line 275, authors claim that the electrostatic repulsion between *p*-C_nTVBs is not the main factor in the superlattice formation. In fig.4 they observed all the possible peaks for *n*=10 in the mixture and for *n*=10, zeta potential is -25 mV. Did the authors check their claim that there are no charging effects on C12E5 cylinders by screening the interactions with salt for all the mixing ratios and *n*? What is the effect of pH?

[Reply]

To identify the effects of electrostatic repulsion, the SAXS measurements with and without charge screening were performed for *p*-C₁₀TVB/C₁₂E₅/water which contains most highly charged particles. As it is shown in the Supplementary Fig. 4, the SAXS intensities of *p*-C₁₀TVB/C₁₂E₅/water (5/45/55 and 10/45/55) in which the electrostatic repulsion was screened by 10 mM NaCl show the same scattering patterns as those without NaCl. Since we

did not observe any charge effects in the sample which contains the most highly charged particles, we didn't performed charged screening experiments for other samples. In the revised manuscript, we clarified that we performed SAXS measurements for the sample which contains most highly charged particles ($n = 10$). (page 17, line 268)

As we answered above for the question about pH value in our experiment, we performed all our experiments at neutral pH (using water purified with a Millipore Direct Q system without any adjustment of pH) and did not aim to investigate the pH dependency of $p\text{-C}_n\text{TVB}/\text{C}_{12}\text{E}_5$ /water system. It is well known that the surface charge of charged particles depends on pH (S. Seal et al, Biomaterials 28, 4600 (2007)). Therefore, identifying the effects of pH on our superlattice formation may be similar to identifying the effects of electrostatic repulsion. Our charge screening experiments essentially answered the pH effects, although some other pH effects may appear. It has been reported that the hexagonal phase of C_{12}E_9 /water (a non-ionic surfactant system which is similar to our C_{12}E_5 /water system) does not show any structural transition depending on pH (G. Kumaraswamy et al, J. Phys. Chem. B, 115, 9059, (2011)). In their study, they performed SAXS measurements of C_{12}E_9 /water (50/50) at different pH (1, 5, 12.8), which show no structural change with pH. Considering this, we expect that the hexagonal phase of our system (C_{12}E_5 /water, 45/55, which is a base system of our study) would not change with pH.

Reviewer #2 (Remarks to the Author):

The manuscript of Lim et. al. describes periodic assemblies from binary 1D micelle-like structures. In my opinion, the authors demonstrate the interesting cases of assembly that are to the best of my knowledge were not discussed before, that is important from fundamental point of view. The experimental work is detailed and analysis seems to be thorough. I can see this manuscript being published in Nature Communication.

The only issue that I want to bring up is that the text of the manuscript is overwhelmed by technical details - it is well written text, but since it has some technical details it is hard to follow the idea of the experiment and the conclusions. Also it will be great if authors will add the comments about the practical value of this study.

[Reply]

We thank the reviewer for very encouraging comments on our work. We agree with the reviewer that our manuscript contains technical details, but it is to describe our experiments, analysis and understanding clearly. We are afraid that removing or simplifying the details may obscure our explanation. As suggested by the reviewer, we added comments about this work's connections to future practical applications in the conclusion. (page 21, line 374)

Reviewer #1 (Remarks to the Author):

I recommend to publish the manuscript in the current form. I understated the difficulties involved in real space measurements using Cryo-TEM.